# A Review of the COVID-19 Mental Health Impact in Post-Conflict Settings: Bridging the Mental Health Gap with Case Exemplars from an Implementation Science Lens

**DOI:** 10.3390/ijerph20116006

**Published:** 2023-05-31

**Authors:** Elizabeth Noble, Deborah Adenikinju, Christina Ruan, Sophia Zuniga, Diksha Thakkar, Carly M. Malburg, Joyce Gyamfi, Temitope Ojo, Farha Islam, Amy Cohen, Lotanna Dike, Chinenye Chukwu, Siphra Tampubolon, Emmanuel Peprah

**Affiliations:** 1Global Health Program, NYU School of Global Public Health, New York, NY 10003, USA; 2Department of Social and Behavioral Sciences, NYU School of Global Public Health, New York, NY 10003, USA

**Keywords:** mental health, COVID-19, post-conflict settings, evidence-based interventions, implementation science, CFIR

## Abstract

The COVID-19 pandemic has further aggravated the burden of mental health and presents an opportunity for public health research to focus on evidence-based interventions appropriate for populations residing in resource-constrained, post-conflict settings. Post-conflict settings have a higher service gap in mental health and fewer protective factors, such as economic and domestic security. Post-conflict settings are defined as locations where open warfare has ended but resulting challenges have remained for years. A strong emphasis on the engagement of diverse stakeholders is needed to arrive at sustainable and scalable solutions to mental health service delivery. This review discusses mental health service delivery gaps in post-conflict settings, highlights the urgency of the matter in the context of the COVID-19 pandemic, and provides recommendations for service gaps from evidence-based case study exemplars with an implementation science lens using the Consolidated Framework for Implementation Research (CFIR) as guide to improving adaptation and uptake.

## 1. Introduction

Living conditions in post-conflict settings coupled with limited resource access due to the COVID-19 pandemic have contributed to increased emotional distress and risk to mental health [1,2]. Post-conflict settings are defined as locations where open warfare has ended but resulting challenges have remained for years in political stability, security, justice, socioeconomic development, and social equity [1]. Although stress leading to increased risk to mental health is ubiquitous in post-conflict settings, some groups within these areas are more vulnerable to the psychological consequences of the pandemic [3]. Vulnerable groups include older adults, individuals with pre-existing medical conditions, health workers, and those who contracted mild to severe COVID-19 infections [2,4]. Further, resource-related constraints often present differently across post-conflict settings as they are often powered by economic, political, societal, and cultural factors [1]. Therefore, tackling these issues are often complex and difficult [5].

Studies show in populations affected by conflict, the rate of increased mental health risks is more than double that of the general population, and in these settings, one out of five people live with a mental health disorder ranging from depression to psychosis. Presented here are the cases of Kosovo, Zimbabwe, and Guatemala which have historically experienced civil war and subsequent civil unrest, disrupted resource allocation, prioritization, and service advancement in mental healthcare [6,7,8]. The impact of conflict on country development and population mental health shows the critical need for implementation of effective evidence-based interventions (EBIs) in mental health service delivery in these settings [1].

Further, preventive measures to decrease the spread of SARS-CoV-2 during the COVID-19 pandemic, such as physical distancing, self-isolation, travel restrictions, and quarantine, have impacted the mental health of people worldwide [9]. Due to these nonpharmaceutical interventions (NPIs), people are engaging less in social interaction and experiencing increased loneliness as they are away from family and friends [10]. In post-conflict settings, stressors related to COVID-19 such as loneliness, anxiety, uncertainty for the future, and government restrictions have greatly affected the population’s overall mental health [3,9,10]. A recent survey conducted in Bangladesh reported a 33% increase in the prevalence of depression and a 5% increase in suicidal ideation as a result of the pandemic [3]. Additionally, economic pressures contribute to many mental health risks, including anxiety and depression [3,9]. Global economic growth is expected to decrease by 8% following the pandemic [3]. A reduction of this size has the potential to push over 100 million people into extreme poverty [3]. Furthermore, due to mandated travel restrictions and self-isolation people are experiencing job loss, increased domestic violence, and substance abuse [10].

The Global Disease Burden Study has found that anxiety, mood, trauma, and stress disorders are all linked to social determinants of health [4]. For instance, economic stress, employment insecurity, social isolation, decreased access to community support, barriers to mental health services, and exacerbated physical health problems impact an individual’s mental health and socioeconomic status [1]. This, in turn, increases mental health risks [3].

Identifying sustainable EBIs to address mental health-related conditions and ongoing measurement of implementation success is crucial to addressing the mental health service gap. Furthermore, understanding the “what”, “why”, and “how” behind the effectiveness of EBIs will ensure appropriate and effective implementation in various contexts [5]. Additionally, all EBIs should attain optimal reach, efficacy, adoption, implementation, and sustainability to achieve maximum success [5]. This review analyzes three case exemplars that demonstrate the successful implementation of EBIs that aid in bridging the mental healthcare gap caused by the COVID-19 pandemic within post-conflict settings, selected via a thorough literature review. Table 1 outlines the three EBIs mapped onto the domains of the Consolidated Framework for Implementation Research (CFIR) [11].

CFIR is a determinant framework with 39 constructs grouped into five main domains [6]. The five domains of CFIR include “Intervention Characteristics”, “Outer Setting”, “Inner Setting”, “Characteristics of Individuals”, and “Process” [11]. CFIR aids researchers in identifying the constructs that are most relevant to their setting to guide the implementation process and ensure positive outcomes [11]. The CFIR framework delivers a standardized structure for building on findings across studies [11]. A systematic review has shown that applying CFIR in post-implementation research is meaningful in linking CFIR constructs (determinants of implementation) to outcomes (implementation or innovation effectiveness) [11]. CFIR was chosen to showcase innovative EBIs for post-conflict, resource-constrained settings, where adaptation and adoption are critical components of successful mental health service delivery in disastrous circumstances exacerbated by the COVID-19 pandemic.

The aim of this review is twofold. First, CFIR demonstrates how these EBIs worked around and within health system constraints to ensure continued access to mental healthcare. Secondly, we argue for the continued implementation of these or similar EBIs in post-conflict settings to ensure that individuals continue to receive mental health care during future disasters. Overall, this review identified the following EBI themes: telehealth, task shifting, and media advocacy.

## 2. Methods

A literature search was conducted to identify articles that implemented EBIs in an innovative or novel way that allowed successful mental health service delivery, despite delivery deterrents caused by the COVID-19 pandemic in post-conflict settings. The search was conducted in the following research databases: PubMed, Cochrane, Embase, Web of Science, and PsycInfo. A search strategy was developed using the following terms and input for each database: “mental health”, “anxiety”, “depression”, “PTSD”, “stress”, “mental disorder”, “trauma”, “COVID”, “inequity”, “post-conflict”, “resource-constrained”, “low resource”, “poverty”, “resource-poor”, “LMIC”, “low-income country”, “developing”, “underserved”, “marginalized”, “psychological”, “wellbeing”. Appendix A details our chosen search terms, the search strategy applied in the databases, and our inclusion and exclusion criteria. Formative research revealed that the selected search terms encapsulated the current literature on our chosen topic. PRISMA guidelines were applied when screening the articles to establish rigor and an unbiased screening review. Our inclusion criteria were set to include mental health studies that occurred within post-conflict, resource-constrained settings defined using the World Bank classification for LMIC, utilized evidence-based methods, and were directly linked to the COVID-19 pandemic [12]. Our exclusion criteria were set to exclude all studies that took place in High-Income Countries (HIC) or countries that are not considered to be a post-conflict setting, as well as lack of focus on the COVID-19 pandemic or mental health services.

A total of 1077 articles were found from all five databases (Appendix A). After removal of duplicates (*n* = 111), 966 articles were screened, each by two researchers in two phases, against the inclusion and exclusion criteria. Phase one of the screening consisted of two researchers applying the screening criteria to the title and abstracts of the 966 articles. A total of 717 articles were excluded after the first phase. Phase two of the screening consisted of two researchers applying the screening criteria to the full text of the remaining 249 articles. Following phase two of the screening, 10 articles were accepted to be considered for this review. In instances where there were conflicts in how the two researchers reviewed the articles, a third corresponding researcher reviewed the article and resolved the conflict by deciding whether the article should move forward.

Appendix A shows the synthesized findings of the 10 articles by listing the research setting, purpose/aim, intervention, the impact of the intervention, and the impact of the COVID-19 pandemic on implementation. After reviewing, we identified the following themes in mental health EBIs: telehealth, task shifting, and media advocacy. Ultimately, we selected to showcase only three of the ten articles to be included in the final analysis for this review as the quality of the three articles was superior to the remaining seven. Furthermore, the remaining seven articles were excluded post-screening as the intervention was not as innovative or as greatly impactful in the COVID-19 pandemic compared to the three chosen articles. The three chosen articles presented noteworthy examples of successful implementation of interventions that bridge the mental health gap caused by the COVID-19 pandemic within post-conflict settings. The case exemplars as presented in Table 1 describe in detail how these EBIs possess strong advantages for different populations and care levels within the scope of CFIR have been successfully implemented in varying contexts and yield improvements in health outcomes [7,8,11]. The cases presented offer detailed characteristics of how each EBI has the potential for high adaptability and uptake in similar settings.

## 3. Case Exemplars

### 3.1. Intervention Case Exemplar—Telehealth

In response to the disruption of in-person services caused by the COVID-19 pandemic, professionals and patients alike were forced to seek digital solutions [6]. Telehealth, the delivery of medical services using electronic means, provides long-distance rehabilitation care for patients. An example of the effectiveness of such an approach can be seen in Kosovo [6]. Given that 93% of the population has access to the internet, the government implemented psychoeducation videos, helplines, and hotlines for psychological first aid [6].

The provisions in Kosovo demonstrated that in a time when face-to-face services were not viable, utilizing technology under minimal supervision from the government allowed citizens to take full advantage of telehealth opportunities [6]. Initial support from a political and financial level, as seen in Kosovo, is critical in the beginning stages of building telehealth interventions [6]. Manuals must be developed to improve the process of establishing hotline crises, as well as the delivery of high-quality services [6]. Nonetheless, the reflection of existing experiences whilst applying them to other crises could very well lead to the duplication of information [6]. The process also fails to take into consideration the post-evaluation of the services itself [6]. However, such limitations can be minimized by implementing short- and long-term training through hands-on application, with regular check-ins [6]. Although time consuming, training is necessary to maintain the momentum of establishing emergency services over a short period of time [6]. Continuous training should also be provided for volunteers in psychological first aid. Correspondingly, more training for online mental health services should be included in psychology programs to prepare future professionals [6]. Moreover, marketing and promoting the availability of these emergency telehealth services, whether through SMS or social media, is critical to long-term success [6]. It was observed that when the helpline was promoted by notifying phone numbers in Kosovo, the number of calls for mental health services increased significantly [6]. Additionally, the promotion of webinars through social media led to an increase in participation [13].

The application of telehealth on a system level can be applied in similar post-conflict settings guided by CFIR as seen in Table 1. As of now, the majority of online psychology programs do not include online mental health training for capacity building and thus, the training of participants should include courses related to digital mental health [6]. Training must be continuous in order to ensure quality improvement over time to address changing needs [6]. Telehealth may also result in decreased stigmatization by creating more trust and access to providers and mental health services remotely [6]. Therefore, utilizing the domains of CFIR, such as intervention characteristics to apply digitized mental healthcare, whilst detailing the various components of the intervention suitable for the implementation context, would result in improved access to mental health services. This strategy is promising for successful intervention adaptation and uptake.

As seen in Kosovo, telehealth interventions are viable and implementable in post-conflict settings with limited physical access to mental healthcare [14]. However, further development and use of digital services in the mental health space are needed, including understanding its evaluation in terms of effectiveness and confidentiality, as legal concerns and skepticism remain challenging [13].

### 3.2. Intervention Case Exemplar—Task Shifting

Task shifting is an EBI that addresses mental health service delivery challenges caused by the COVID-19 pandemic within post-conflict settings by decentralizing mental healthcare [14]. An example of task shifting is training community health workers to provide mental health services [7]. This enables community members to feel comfort in having local members provide guidance with a sense of solidarity and contextual understanding [7]. A study in Zimbabwe showed the benefits of having local community members trained to improve mental healthcare [7]. Grandmothers were trained to provide mental health assessments and problem-solving therapy to community members; they were chosen to provide these services due to their strong ties to the community in respect and trust [7]. This case showed the success of task shifting, allowing a well-trusted and trained community member to provide mental healthcare to other community members [7]. Task shifting within communities will not only provide a stronger sense of safety, but also allow community members to help health workers share the burden of mental healthcare in post-conflict settings [7].

The local level intervention of task shifting can be applied using CFIR to decentralize mental healthcare with increased promotion of community services [15]. All key aspects of interventions within CFIR must be considered, such as peer mental health workers’ source, adaptability, program design, and cost. The characteristics of this intervention include having community-based peer mentors helping with service delivery in post-conflict settings [15]. This education may be provided through a public–private partnership between NGOs, educational institutions, and community groups to increase reach in underserved populations [15]. Peer mentors would be trained by medical professionals on how to be effective and successful mental health advocates [15]. Thus, community members can become peer mental health mentors, and medical professionals may reprioritize some of this time for overall medical management [15]. The study identified limitations in patient follow-up attendance, documentation of services provided, and lack of available referral resources—revealing further improvement needs in mental health infrastructure support to achieve better outcomes [15].

Normalizing conversations on mental health will increase community members’ awareness and improve their comfort level in discussing this challenging topic [8,15]. With the sizable cultural stigma around mental health, having community members who work as peer mentors will foster a healthier thought process surrounding mental health, such as further acceptance [3,15]. Utilizing CFIR as a guide (Table 1), community-based peer mentors have shown successful outcomes where adaptation and adoption both increase due to having more awareness of the importance of community-led mental health service delivery and support within post-conflict settings. This framework will allow similar settings to follow the steps of implementing local mental health interventions effectively with successful outcomes, in addition to improving service delivery infrastructure [3].

### 3.3. Intervention Case Exemplar—Media Advocacy for Behavior Change

The case study from Guatemala highlights media advocacy as a promising mental health intervention to scale up [8]. This intervention utilizes media messaging to promote mental health awareness, help-seeking behavior, and self-care education [8]. Individuals are able to recognize signs of mental health distress while receiving messaging that normalizes and validates the individual’s experience, and promotes community mental health resources [8]. Mental health education consisted of Cognitive Behavioral Therapy (CBT) activities for mindfulness and emotional regulation at home [8]. CBT is most effective in the setting of the COVID-19 pandemic as CBT emphasizes the individual as an active agent in treating oneself, giving back control in an uncontrollable situation [8]. Media advocacy has the potential to be a most effective mental health intervention as it has high reach over a short timeframe, while addressing the most fundamental barriers of mental health stigma and access to care due to geographic distance or cost [8]. Other than the effectiveness of this intervention for improving mental health care in post-conflict settings, it is important to note that this approach leads to successful implementation through community ownership and capacity building [8].

In the Guatemala media advocacy case, this intervention was found particularly effective in the emergent situation of the COVID-19 pandemic [8]. Additions to this approach must include access to community resources, complementary mental health services, promotion of clear behavior change recommendations, updated service delivery over time, and a multilevel intervention for sustainability [8]. It is critical to note that media advocacy should be seen as complementary to other mental health interventions, as internet and media access may be a barrier [8]. Additionally, since the outcome of media advocacy is help-seeking behavior change, there must be mental health services available to refer the audience [8]. Interventions that address help-seeking behavior change have been found critical in the pandemic due to significant help-seeking barriers such as reduced income, commute challenges, increased service cost, and fear of COVID-19 infection or stigmatization [16].

The Guatemala media advocacy case further identified three factors of implementation that led to high success [8]. This includes community participation in program design, utilizing culturally appropriate content, and adaptation to changing socio-behavioral contexts [8]. Community participation increases adoption by identifying relevant needs for targeted messaging in a ground-up approach of modification according to local values, attitudes, and beliefs around mental health [8]. This approach also leverages social network resources, such as mental healthcare services at churches and schools for increased public access [8]. This is most effective in creating the desired outcome of help-seeking behavior change, while this messaging must also adapt over time per the changing socio-behavioral context [8]. For example, at the start of the pandemic, the predominant mental health complaint was rising anxiety due to many unknown concerns. As the pandemic continued, this changed to rising depression from isolation, economic decline, disrupted services, and lost loved ones. As the predominant mental health complaints changed, so did the messaging approach to ensure the community was always receiving the most relevant resources [8]. This media advocacy intervention is mapped out for successful implementation utilizing CFIR in Table 1.

Limitations of this intervention as noted by researchers include significant time constraints in implementation and evaluation due to the urgency of the matter [8]. Consequently, researchers were unable to build a significant process of evaluation to examine the long-term effects of the campaign [8]. Nonetheless, the underlying agreement is that increased exposure from campaigns that (1) take into account the ease of access to resources to drive behavioral changes, (2) promote clear recommendations, and (3) are adaptable and ultimately have the best chance of being successful [8]. Similarly, to address the emerging issue of an “infodemic” or messaging fatigue, it is vital that media advocacy addresses misinformation with trusted sources to ensure greater uptake among the population [8].

Health communication models contain factors for effectiveness such as media mode or messaging characteristics, that do not necessarily apply the same way in settings where mental health stigma is high [8]. In the Guatemala media advocacy case, cultural values (e.g., keeping mental health issues to one’s self to avoid shaming the family) serve as a barrier to help-seeking behavior [8]. This calls for messaging that first addresses the culturally prevailing attitude in order to have the most impact on mental healthcare adoption [8]. Key to this intervention is partnering with local organizations and trusted supporters of the community such as civil society groups, religious and community health leaders [8]. These partners will assist in the messaging design, dissemination, and implementation of mental health service delivery [8].

This showcase of media advocacy for behavior change serves as a model that may be duplicated in similar settings for the successful implementation of mental health interventions that are well adapted and highly adopted [8]. This case also highlights the limited number of mental health campaigns in post-conflict settings, with even fewer studies on the actual implementation and effectiveness of these campaigns in high-risk, underserved communities [8]. Overall, the limited mental health resources present a significant barrier to population-level change in mental healthcare in post-conflict settings—where a focus on highly innovative EBIs must be implemented [8].

## 4. Conclusions

The COVID-19 pandemic is ongoing and continues to bring countless challenges to every country around the world. Post-conflict settings have been especially susceptible to poor outcomes related to the COVID-19 pandemic because of limited resources and insufficient infrastructure [4,9,16]. Due to the continually evolving pandemic, evidence is limited, but the current literature suggests that individuals with increased mental health risks residing in post-conflict settings have been particularly negatively impacted during the COVID-19 pandemic [6,7,8,14,17,18,19]. Countries have reported increased rates of depression, anxiety, and interruptions in administering standard-of-care interventions for mental health [3]. This is a result of government restrictions, resource shortages, and supply chain issues [1,3,4,15]. However, healthcare is adaptive and three non-traditional EBIs have emerged to address these unprecedented obstacles: telehealth, task shifting, and media advocacy [6,7,8].

Telehealth interventions allow mental health services to be more accessible and reach a larger population than before the pandemic [6]. Task shifting is a cost-effective intervention that is community led and increases mental health participation and trust [7]. Media advocacy is also a cost-effective intervention that better addresses mental health triggers, limits messaging fatigue, is a good complement to standard mental health interventions, and ultimately has the greatest reach [8]. Furthermore, all three interventions can be easily adapted to the different needs, cultural stigmas, and social determinants that are present in diverse settings and contexts.

## 5. Future Directions

Finally, as the pandemic continues to present challenges, it is difficult to define every way that it has affected or will affect global mental health. A substantial opportunity to continue scaling up and adapting mental health interventions, in addition to other NPIs in this ever-changing climate to improve outcomes for populations. It is important to note that with compounding risk factors of extreme stress, high disease prevalence, and poverty, individuals are extremely vulnerable to the negative consequences of poor mental wellbeing. In other words, such unprecedented circumstances may lead to a poor pre-existing mood that can discourage population participation. Consequently, interventions may be less effective in uptake.

Limitations identified by this review excluded studies on adolescents and populations with psychological illness. There is a significant lack of current research and interventions specific to these populations in post-conflict settings that require future efforts in closing this mental health gap.

Additionally, researchers should emphasize the negative repercussions associated with poor mental health, which ultimately may impact an individual’s socioeconomic status, and the community’s wellbeing. Ensuring mental health service infrastructures are resilient and prepared for disastrous events requires further innovative development, study, and continuous quality improvement by practitioners to minimize service disruption. Policymakers should not underestimate the power of social capital and should seek to significantly improve mental health services, whilst taking into consideration the cultural values of the intended target population. Partnering with local stakeholders and trusted community leaders, such as religious organizations and civil society organizations, will be instrumental in changing the beliefs, attitudes, and practices surrounding mental health. Creating a culture of enacting mental health as a human right will require a societal multidisciplinary approach. The way forward must include decentralizing mental healthcare with local service capacity building, creating behavior change interventions that destigmatize mental healthcare for greater acceptability, and prioritizing the cultural adaptation of well-adopted mental health services.

## Figures and Tables

**Table 1 ijerph-20-06006-t001:** CFIR Intervention Analysis.

Case Exemplars	Intervention Characteristics	Outer Setting	Inner Setting	Individual Characteristics	Implementation Process
Kosovo: Telehealth [6]	- system level online and mobile psychological first aid services - development of psychoeducational videos and webinars	- policy change - government leadership - economic measures - comprehensive training - media reach and engagement	- training incentives - community leadership	- decrease mental health stigma - increase mental health literature, information, and evidence-based practice - mental health service capacity building	- easy to access support platform and presentation of psychoeducational material - complementary to other mental health interventions - increased mental health coverage
Zimbabwe: Task Shifting [7]	- local level peer mentor program - CHW training - community based education and support - increased access to local mental health services	- health system availability of mental health trainers - health system integration - local infrastructure	- community leadership - local employment incentive	- decrease mental health burden within communities - decrease mental health shame within a cultural setting - increase mental health comfort with trusted and relatable caregivers	- community mental health peer mentor training - local empowerment - cost effective alternative to physician care - increased mental health coverage - decentralized mental health care - intervention evaluation
Guatemala: Media Advocacy [8]	- individual level behavior change - promote mental health awareness and help-seeking behaviors - combat misinformation - self-care education - trusted health leaders and influencers sharing mental health stories	- cultural stigma - local mental health service resources - media reach and engagement - high-radio reach in settings where social media penetration is low	- social network - stakeholder engagement - community leadership	- decrease mental health guilt or shame - mental health sensitization - increasing mental health dialogue - appropriate fit for young adults	- participatory program design - cost-effective access to large population over wide distance - ground up identification of stressors for target population - complementary to other mental health interventions - adaptation for messaging fatigue - intervention evaluation

## Data Availability

The data presented in this study are openly available in PubMed, Cochrane, Embase, Web of Science, and PsycInfo online databases, reference numbers [7,8,11].

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
