# Peer review of "A Review of the COVID-19 Mental Health Impact in Post-Conflict Settings: Bridging the Mental Health Gap with Case Exemplars from an Implementation Science Lens"

_ijerph, 2023, doi:10.3390/ijerph20116006_

Round 1
Reviewer 1 Report
This manuscript provides a good overview of the impact of COVID-19 on mental health in post-conflict settings and presents case exemplars of effective evidence-based interventions (EBIs) that can bridge the mental health gap. However, I have some suggestions that might help clarify or develop the quality of this manuscript.
1) The introduction can be strengthened by providing more context and background information on the mental health impact of conflict and post-conflict settings, as well as the impact of COVID-19 on mental health in general.
2) The research would benefit from a more systematic literature review, including a clear search strategy, selection criteria, and quality assessment of the studies included.
3) Methods
3.1) To make this research easier to understand, I suggest that the authors add a diagram showing the article selection process.
3.2) Line 80: Clarify the criteria for selecting the three articles: The rationale for selecting three out of the ten articles for the final analysis could be better explained. What criteria were used to select these three articles? How were they deemed superior to the remaining seven? Providing more details on this process would make the selection process more transparent.
3.3) Provide more information on the synthesis process: It would be helpful to provide more information on how the findings of the ten articles were synthesized. What methods were used to synthesize the information? Were there any discrepancies between the two researchers who screened the articles? How were these discrepancies resolved?
3.4) Consider a more detailed analysis of the selected articles: While the three selected articles are described as presenting noteworthy examples of successful implementation of interventions, it would be useful to provide a more detailed analysis of these articles. What specific interventions were used? What were the outcomes of these interventions? Were there any limitations or challenges to the interventions? Providing this level of detail would enhance the quality of the analysis.
4) The discussion could be expanded to provide more in-depth analysis of the findings and to compare and contrast the case exemplars presented. The authors could also consider discussing the limitations and challenges of implementing EBIs in post-conflict settings, as well as future research directions.
5) The authors could consider including more practical recommendations for policymakers, practitioners, and researchers on how to effectively implement EBIs in post-conflict settings, taking into account the unique challenges posed by COVID-19.
6) The writing could be improved by using clearer language, avoiding repetition, and organizing the content into more cohesive sections.
7) Additionally, the authors should provide more explicit links between the case exemplars presented and the theoretical framework used (CFIR).
Reviewer 2 Report
Overall a nice paper, well-written and think it would be of interest to readers. Some comments below.
Abstract: If you have not exceeded the word count, it would be useful to include either the definition (shortened) of post-conflict settings or the examples you include in the paper.
Introduction: Expand on the explanation of the CFIR framework and consider including examples of it being used in other research.
Methods: At present, the section does not contain sufficient information to allow replicability. You state that the search was conducted following the PRISMA guidelines, there is no checklist attached to suggest that you have done this. Also, there is no PRISMA flow diagram to display how the articles were screened. No details on how you excluded papers based on the quality of the studies.
